# Which Clusters of Metabolic Syndrome Are the Most Associated with Serum Uric Acid?

**DOI:** 10.3390/medicina58020297

**Published:** 2022-02-16

**Authors:** Jurgita Mikolaitytė, Jolita Badarienė, Roma Puronaitė, Alma Čypienė, Irma Rutkauskienė, Jolanta Dadonienė, Aleksandras Laucevičius

**Affiliations:** 1State Research Institute Centre for Innovative Medicine, LT-08406 Vilnius, Lithuania; jolita.badariene@santa.lt (J.B.); alma.cypiene@santa.lt (A.Č.); aleksandras.laucevicius@santa.lt (A.L.); 2Clinic of Cardiac and Vascular Diseases, Faculty of Medicine, Institute of Clinical Medicine, Vilnius University, LT-03101 Vilnius, Lithuania; roma.puronaite@santa.lt; 3Faculty of Mathematics and Informatics, Institute of Data Science and Digital Technologies, Vilnius University, LT-03225 Vilnius, Lithuania; 4Centre of Cardiology and Angiology, Vilnius University Hospital Santaros Klinikos, LT-08661 Vilnius, Lithuania; irma.rutkauskiene@santa.lt; 5Clinic of Reumatology, Faculty of Medicine, Institute of Clinical Medicine, Vilnius University, LT-03101 Vilnius, Lithuania; jolanta.dadoniene@santa.lt

**Keywords:** metabolic syndrome, clusters of cardiovascular risk factors, hyperuricaemia, uric acid

## Abstract

*Background and Objectives*: Metabolic syndrome is defined as three or more of five components; therefore, there are 16 possible different clusters of metabolic risk factors that are under one diagnosis of metabolic syndrome. In this study, we evaluated the different clusters of metabolic syndrome (MetS) across serum uric acid (SUA) quartiles and analyzed the association of these clusters with SUA levels, respectively, in both men and women. *Materials and Methods*: A total of 606 subjects were recruited to a cross-sectional study from the ongoing Lithuanian High Cardiovascular Risk primary prevention program (LitHiR). All of the study subjects were diagnosed with MetS (according to the 2005 National Cholesterol Education Program Adult Treatment Panel III MetS definition). *Results*: In the middle-aged population of patients with MetS living in Lithuania, a high proportion of hyperuricaemia was detected—35.5% (95% Cl 31.7% to 39.4%). For women possessing all five MetS components, the chances of having hyperuricaemia are 2.807 higher than for women with three risk factors (*p* < 0.001). However, men do not have a statistically significantly higher chance of having hyperuricaemia, depending on the number of MetS components in our population. Using multivariable models, the statistically significant chance of having hyperuricaemia was observed only in women possessing all five MetS components (OR = 2.386, *p* < 0.0001), compared to any other of 15 MetS clusters. After adjustment for age and sex, the chance of having hyperuricaemia for individuals with the cluster of all five MetS components, compared to any other of 15 MetS clusters, remained (OR = 1.982, *p* = 0.001). Also, a lower probability (OR = 0.653, *p* = 0.039) of having hyperuricaemia was observed for individuals having the combination of abnormal plasma glucose, blood pressure, and waist circumference. *Conclusions*: Patients with the clustering of all five metabolic syndrome components are at higher risk for having hyperuricaemia than patients with any other combination of MetS clusters. This risk is even higher for women. It could be beneficial for patients presented with all five MetS components to be screened for SUA concentration in the primary CVD prevention program.

## 1. Introduction

Increasing attention has been paid to the relationship between serum uric acid (SUA) and metabolic syndrome (MetS). Although SUA is not one of the criteria for diagnosing MetS, it is considered to be correlated with MetS, as both are closely linked with cardiovascular diseases and type 2 diabetes [1,2,3]. MetS prevalence among subjects with hyperuricaemia is very high [4]. Recent epidemiological studies have demonstrated an association between SUA and MetS, and their components in different populations [5,6,7,8,9,10,11,12]. However, the correlation between SUA and MetS and its components varies, depending on subjects’ age, sex, ethnicity, workplace, and other factors [5,6,7,8,9,10,11,12].

Metabolic syndrome is defined as three or more of five components; therefore, there are 16 possible different clusters of metabolic risk factors that are under one diagnosis of metabolic syndrome. Those different MetS clusters may present different pathophysiology, consequences, and treatment options, depending on which MetS components are present [13]. The prevalence of the different MetS clusters differ across age, sex, ethnicity and culture [14]. Thus, MetS used as a whole may mask important differences in assessing health and mortality risk. Indeed, studies that have compared MetS clusters for their ability to predict mortality have demonstrated variations in mortality risk among different MetS clusters or components [13,15,16,17].

In this study, we evaluated the different clusters of MetS components across SUA quartiles and analyzed the association of these clusters with SUA levels, respectively, in both men and women.

## 2. Materials and Methods

A total of 606 subjects were recruited to a cross-sectional study from the ongoing Lithuanian High Cardiovascular Risk primary prevention program (LitHiR) [18]. The study is in line with the principles outlined in the Declaration of Helsinki and approved by the Vilnius Regional Ethics Committee for Biomedical Research, Vilnius University (Protocol No. 158200-18/4-1006-521, Approval date 3 April 2018). Written informed consent was obtained from each participant. The inclusion criteria were as follows: the age for men was between 40 and 55 years, for women, 50 and 65 years. MetS was diagnosed according to the 2005 National Cholesterol Education Program Adult Treatment Panel III (NCEP ATP III) definition. The different age ranges for men and women were chosen in accordance with European Society of Cardiology Guidelines on cardiovascular disease prevention in clinical practice, which state that 60-year-old women have similar risk for CVD as 50-year-old men [19]. Patients with previously diagnosed cardiovascular, cerebrovascular or peripheral artery disease and end-stage oncological disease were not included.

All patients underwent a physical examination, including the measurement of height, weight, waist circumference, blood pressure, and blood collection. Height and weight were measured in light clothing without shoes. Body mass index (BMI) was calculated as body weight (kilograms) divided by the square of height (square meters). Waist circumference was measured by placing a flexible tape measure around the waist at the navel level. Blood pressure was measured using a standard sphygmomanometer in a sitting position after a period of 10 min of rest. Blood samples were obtained after 8–12 h of overnight fasting. The following laboratory parameters were determined: SUA, lipid panel, and plasma glucose. The hyperuricaemia was defined when the SUA levels in blood serum exceeded 357 µmol/L in women and 428 µmoL/L in men. 

All analyses were performed using STATISTICA (version 10) and R (version 3.6.1). Descriptive statistics were computed for demographic information. Data were expressed as mean ± standard deviation for continuous variables and as absolute and percentage values for categorical variables. Continuous variables were analyzed using Student’s t-test or Mann–Whitney U test, categorical variables with Pearson’s chi-square test. The normality of data was a test using Shapiro–Wilk test. Normally distributed data were analyzed using one-way ANOVA, non-normally distributed data using Kruskal–Wallis tests. Logistic regression was performed to assess risk factors for hyperuricaemia. Odds ratios with 95% confidence intervals were calculated from multivariable models adjusted for age and sex. For all tests, *p*-values less than 0.05 were considered statistically significant.

## 3. Results

### 3.1. Baseline Characteristics 

A total of 606 participants, with a mean age of 53.78 ± 6.60 years, were included in the study. Of these, 220 of them were men (36.3%) and 386 were women (63.7%). The prevalence rate of hyperuricaemia was 35.5% (95% Cl 31.7% to 39.4%) among all participants, with 34.1% (95% Cl 27.9% to 40.8%) among men and 36.3% (95% Cl 31.5% to 41.3%) among women. Our study population was divided into SUA quartiles by gender. We found that with the increasing SUA concentration, the participants had more elevated values of weight, body mass index (BMI), and waist circumference (WC), in both men and women (*p* < 0.001). Furthermore, there were significant differences among the groups of quartiles in triglycerides (TG), systolic blood pressure (SBP), and the number of MetS components in both genders (Table 1). Men had higher TG in the first quartile than in the other quartiles (*p* = 0.017). The differences in men and women were observed in such variables as age, plasma glucose, and high-density lipoprotein cholesterol (HDL-C). We observed that younger men had higher concentrations of SUA (*p* < 0.001). Plasma glucose in the male group was significantly higher in the first quartile compared to the others (*p* = 0.039). On the other hand, women showed a decrease in HDL-C from the first to the fourth quartile (*p* = 0.004). More detailed results are presented in Table 1.

### 3.2. Prevalence of Major Cardiovascular Disease Risk Factors in the Study Population

Our studied population had a high prevalence of dyslipidaemia (men, 98.2%; women, 96.9%) and primary arterial hypertension (men, 90.5%; women, 93.0%). Therefore, there were no significant differences among quartiles, except men, where primary arterial hypertension prevalence was 100% in the fourth quartile (*p* = 0.034) (Figure 1).

Women had a higher prevalence of diabetes than men (24.4% vs. 17.3%). However, statistically significant differences between quartiles were observed in men (*p* = 0.028). Interestingly, the ratio in men in the first quartile was the highest (Figure 1).

The prevalence of obesity in men was 69.6%, and in women, 64.5%. The gradual growth from the first to the fourth quartiles in both genders was observed (*p* = 0.005 in men and *p* < 0.001 in women) (Figure 1).

### 3.3. Distribution of Individual MetS Components

Analyzing the proportion of risk numbers in MetS gathered in each quartile, five risk factors were found in the upper quartile compared to the lower quartiles (men *p* = 0.019, women *p* < 0.001). Only 18.2% of men and 26.5% of women had three risk factors in the upper quartile (Figure 2).

The most common three individual risk components for both men and women were blood pressure, WC, and plasma glucose. The fourth MetS risk component in men was TG, and in women, TG and HDL-C, almost equally (62.6% and 64.0%, respectively) (Figure 3).

Having analysed the individual components of MetS by SUA quartiles, it can be stated that from all five components, only triglycerides (*p* = 0.031) in men and HDL-C (*p* = 0.015) in women, as MetS components, were statistically significantly more prevalent as SUA quartiles increased. 

### 3.4. Prevalence of Clusters in MetS Components 

Gender differences were also observed, even by analyzing the distribution of MetS clusters according to only the number of MetS components. The clusters of four MetS components were the most common among men (41.4%), and clusters of three MetS components among women (40.7%). The clusters of all five MetS components were presented in both genders similarly (men, 23.6%; women, 23.3%).

In Table 2, the distribution of all 16 different clusters of MetS is presented in men and women. The most common combination for men consisted of all five risk factors (increased plasma glucose, TG, blood pressure, WC and decreased HDL-C)—23.6% for women—a combination whose variables were increased plasma glucose, blood pressure, and waist circumference (28.8%). The least common clusters were plasma glucose, TG, and WC or HDL-C for men, and HDL-C, TG, and blood pressure for women.

### 3.5. Hyperuricaemia and Clusters of MetS Components

Statistically significant higher chances to have hyperuricaemia, depending on the number of Mets components in our population, were observed only in women. Women possessing all five MetS components have 2.807 times higher chance of developing hyperuricaemia than women with only three MetS components (*p* < 0.001). 

Furthermore, to analyze the chances of developing hyperuricaemia among different clusters of MetS, we calculated the odds ratios for the studied population using multivariable models. The statistically significant chance to have hyperuricaemia was observed only in women with the cluster of all five MetS components (OR = 2.386, *p* < 0.0001), compared to any other of 15 different clusters of MetS (Figure 4).

After adjusting the results for age and gender, the chances of developing hyperuricaemia were 1.982 times higher (*p* = 0.001) for individuals with the cluster of all five MetS components, compared to any other of 15 MetS clusters. The lower probability (OR = 0.653, *p* = 0.039) of developing hyperuricaemia was for individuals having the combination of abnormal plasma glucose, blood pressure, and WC. More details are provided in Table 3, where clusters of MetS with an overall prevalence of ≥1.5% are shown. 

## 4. Discussion

Our study provides several important findings. In the middle-aged population of patients with MetS living in Lithuania, a high proportion of hyperuricaemia was detected—35.5% (95% Cl 31.7% to 39.4%). Though we do not have the epidemiological data on hyperuricaemia in the general Lithuanian population, based on the data of other countries (where the prevalence of hyperuricaemia among patients aged 40–59 years is 18.7% in the United States [20], 20.2% in Ireland [21]), we can sustain our claim. The fact that the prevalence of hyperuricaemia increases with age is well established. The prevalence of hyperuricaemia is highest among individuals aged 80 years or older (27.8% in the United States [20], 43% in Ireland [21]). One of the main reasons for this is the use of polypharmacy for multiple comorbidities [22]. 

Women over 50 years of age have an increased risk of developing hyperuricaemia, which is possibly related to decreased estrogen levels [23]. Eswar et al. argued that age is a better statistical explanation for the higher prevalence of hyperuricaemia among older women than menopause status [24]. Our results showed that there were no age-related differences in SUA quartiles in our study group, presumably because all women were at the age of menopause and similarly had no major comorbidities (exclusion criteria for subjects in our study were established CVD).

In our male subjects, who were 40–55 years of age, we observed that younger men had higher concentrations of SUA. Data from other researchers also show that men tend to have a higher rate of SUA at a younger age [25]. L. You et al. suggested that the different concentrations of SUA at a different age for men might be because of the higher level of alcohol consumption and lack of physical activity at the younger age [23]. According to the data in the “Health of the Lithuanian population”, alcohol consumption decreases with age, among both men and women [26]. Though alcohol consumption was not monitored in our study, it could possibly explain the SUA distribution related to age among men.

We found significant positive associations between SUA levels and weight, BMI, WC, and obesity in both genders. These results are in line with many other studies. Though the association between major CVD risk factors, such as hypertension, dyslipideamia, and the levels of SUA, was expected, only men showed a significant increase in hypertension in the fourth quartile of SUA. It should be borne in mind that the majority of the subjects had dyslipideamia (men, 98.2%; women, 96.9%) and hypertension (men, 90.5%; women, 93.0%).

A few differences were observed among men and women, concerning SUA levels and HDL-C, TG, plasma glucose, and diagnosis of diabetes. An inverse association between HDL-C and SUA levels was shown by several studies [27,28]. It seems to be mediated by insulin resistance [29]. However, men in our study did not show any correlation between SUA quartiles and HDL-C level. Low HDL-C was the least prevalent MetS component in men compared to the other four components. Men, in the first SUA quartile, had higher levels of TG, and plasma glucose, and were diagnosed with diabetes. The inverse association of SUA with diabetes mellitus was reported by M. Tao et al. [30]. They speculated that this phenomenon is probably due to the presence of a high level of blood glucose, which promotes renal excretion of SUA. N. Hairong et al. found that individuals with a new diagnosis of diabetes at the baseline had higher SUA levels than during the follow-up, when the diagnosis of diabetes was made, but this was not observed for postmenopausal women [31]. This fact is in line with our findings. 

L. Kuan-Chia et al. found that the increase in SUA levels during the seven year follow-up period was correlated with subsequent diabetes only among hyperuricaemic women. Moreover, a relatively higher incidence of diabetes was found in postmenopausal hyperuricaemic women after a 7-year follow-up [32]. One of the explanations of the sex differences in the prevalence of hyperuricaemia between diabetes and non-diabetes subjects, as theorized by S.Higa et al., could be the flexion point of the glucose concentration [33]. There is evidence that mean SUA increases with an increasing glucose concentration up to 7.0 mmol/L in men and 9.0 mmol/L in women, and thereafter, increasing glucose values are accompanied by a decrease in SUA [34]. This might be the reason why no statistically significant differences among SUA quartiles, in plasma glucose and diagnosed diabetes, were observed in women in our study.

Metabolic syndrome is a general term given to a clustering of cardiometabolic risk factors that may consist of different phenotype combinations, whose occurrence differs at an individual level. There were two of the most common clusters of MetS for our studied population, as follows: a combination of elevated blood pressure, waist circumference, and plasma glucose (24.42%), and clustering of all five MetS components (23.4%). The prevalence of MetS clusters differs greatly among populations in different countries. A. Scuteri et al. analyzed the differences between European countries and found that all the five MetS components had a greater prevalence in Lithuania (13.0%), and a considerably lower prevalence in the Italian cohorts (2–3%) [14]. The cluster of blood pressure, waist circumference, and plasma glucose had the highest occurrence in Southern Europe (Italy, Spain, and Portugal, with 31.4%, 18.4%, and 17.1%, respectively) and Belgium (20.4%), and the lowest occurrence in Northern Europe (Germany, Sweden, and Lithuania, with 7.6%, 9.4%, and 9.6%, respectively) [14]. The area of concern from this data is that the prevalence of the MetS cluster of all five components, compared with earlier data, has grown from 13.0% to 23.4% in Lithuania. These data also indicate the changes in MetS manifestation through time, as the cluster of blood pressure, waist circumference, and plasma glucose increased from 9.6% to 24.42%, and is now the most prevalent combination of MetS in middle-aged Lithuanians.

Many studies have already stated that SUA levels are higher according to the number of MetS components [35]. This corroborates the results observed in our research, since subjects with the presence of all five MetS components showed higher SUA levels, compared to those with three components. Women showed 2.807 times the risk of hyperuricaemia compared to three and five risk factors. However, men did not have statistically significantly higher chances of developing hyperuricaemia depending on the number of Mets components. Moreover, we found that women have 2.386 times the risk of hyperuricaemia, possessing the cluster of all five MetS components, compared to any of the rest of the 15 MetS clusters. In this light, greater attention should be paid to women with all five MetS components, when choosing appropriate treatment and dietary recommendations. After adjustments for age and sex, the chance of developing hyperuricaemia for individuals with the cluster of all five MetS components, compared to any other of the 15 MetS clusters, remained (OR = 1.982, *p* = 0.001). Furthermore, a lower probability (OR = 0.653, *p* = 0.039) of developing hyperuricaemia was observed in individuals having the combination of abnormal plasma glucose, blood pressure, and waist circumference. Both of these clusters (almost equally) were the most prevalent in our population.

The value of hyperuricaemia as a risk factor for cardiovascular mortality was investigated in Finland [36]. The total mortality of hyperuricaemic men and women, over five years, was significantly higher than the mortality of normo-uraemics. Significantly increased cardiovascular mortality, however, was observed only in hyperuricaemic women without known heart disease, in the follow-up period, between 5 and 12 years. A systematic review and dose-response meta-analysis, by Li et al., provided further evidence that hyperuricaemia may increase the risk of CHD events, particularly CHD mortality in females [37]. Therefore, SUA should be assessed regularly, at least in women with all five MetS components, and if the levels are high, hypouricemic agents should be prescribed accordingly, to improve the cardiometabolic profile of the patient.

## 5. Limitations

High consumption of high fructose corn syrup, alcohol, select dietary lifestyles, and use of diuretics is associated with higher SUA levels. The limitation in this study is that we did not evaluate the intake of SUA with food and alcohol. Furthermore, we have not included data on the use of diuretics and beta-blockers. The risk of hyperuricaemia in patients with metabolic syndrome may be influenced by the presence of fatty liver [38]. However, we did not collect data about the presence of fatty liver, or aminotransferase levels.

## 6. Conclusions

Our findings provide evidence for a high prevalence of hyperuricaemia among middle-aged MetS patients in Lithuania. Furthermore, patients with the clustering of all five metabolic syndrome components are at higher risk of developing hyperuricaemia than patients with any other combination of MetS clusters. This risk is even higher for women. It could be beneficial for patients presented with all five MetS components to be screened for SUA concentration in the primary CVD prevention program.

## Figures and Tables

**Figure 1 medicina-58-00297-f001:**
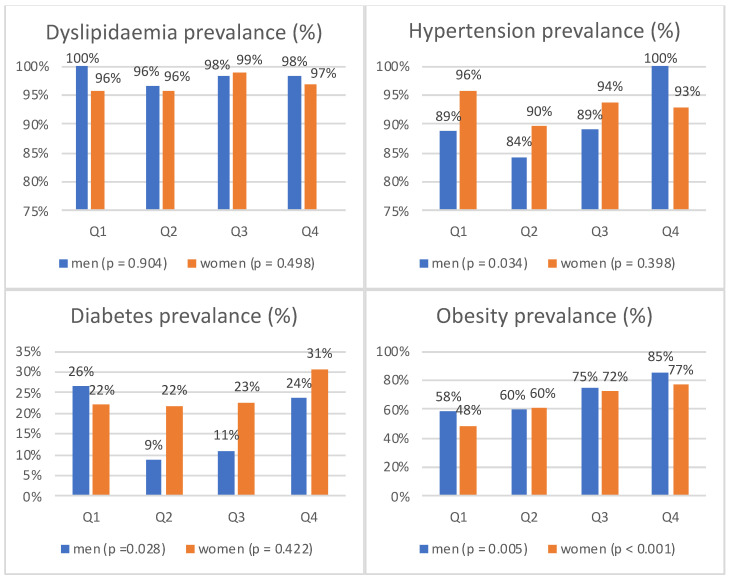
Prevalence of major cardiovascular disease risk factors in different levels of serum uric acid.

**Figure 2 medicina-58-00297-f002:**
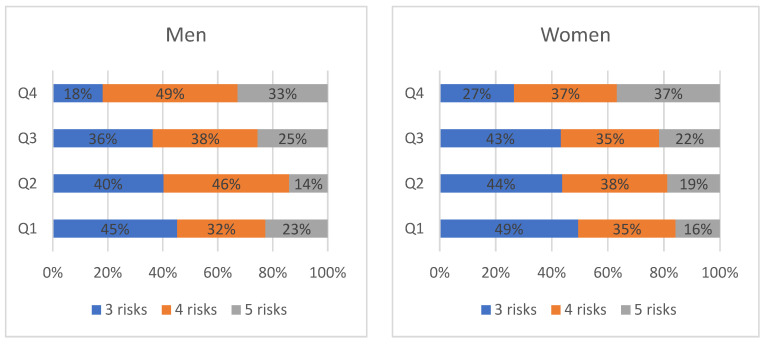
Proportions of metabolic syndrome risk number gathered in each quartile (%) for men and women.

**Figure 3 medicina-58-00297-f003:**
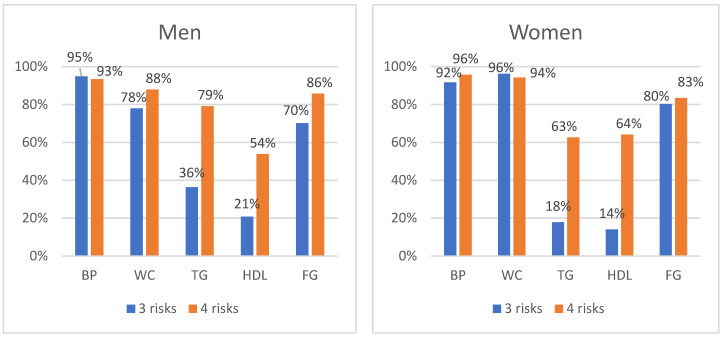
Metabolic syndrome components distribution according to the number of clusters (%) in men and women. BP—blood pressure; WC—waist circumference; TG—triglycerides; HDL-C—high-density lipoprotein cholesterol; PG—plasma glucose.

**Figure 4 medicina-58-00297-f004:**
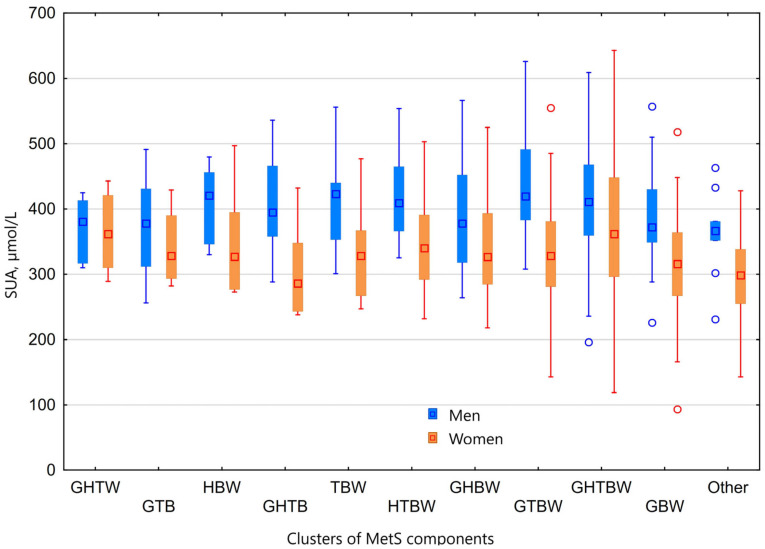
Clusters of MetS components and their association with SUA. SUA—serum uric acid; MetS—metabolic syndrome; G—elevated plasma glucose; H—low high density lipoprotein cholesterol; T—high triglycerides; B—elevated blood pressure; W—increased waist circumference.

**Table 1 medicina-58-00297-t001:** Clinical characteristics of participants in quartiles of serum uric acid level.

	Women (*n* = 386)					Men (*n* = 220)				
	Q1 <283 μmol/L	Q2 283–329 μmol/L	Q3 329–385 μmol/L	Q4 ≥385 μmol/L	*p*	Q1 <353 μmol/L	Q2 353–398 μmol/L	Q3 398–460 μmol/L	Q4 ≥460 μmol/L	*p*
*n*	95	96	97	98		53	57	55	55	
SUA, μmol/L	242.35 ± 36.05	304.79 ± 14.85	354.67 ± 16.21	443.41 ± 55.96	<0.001	312.06 ± 34.50	374.70 ± 12.60	423.98 ± 17.28	511.45 ± 42.73	<0.001
Age, years	57.89 ± 3.95	57.13 ± 4.47	57.90 ± 4.27	57.66 ± 4.01	0.577	48.36 ± 3.67	47.84 ± 4.20	46.22 ± 4.39	45.62 ± 3.75	0.001
Weight, kg	80.62 ± 11.84	85.55 ± 13.63	89.04 ± 14.14	92.31 ±14.20	<0.001	98.09 ±10.21	101.37 ± 13.32	103.36 ± 10.34	109.80 ± 12.19	<0.001
BMI, kg/m^2^	30.28 ± 4.47	31.51 ± 4.34	33.22 ± 4.94	34.06 ± 4.82	<0.001	30.86 ± 3.00	31.08 ± 3.31	32.03 ± 3.25	33.60 ± 3.89	<0.001
WC, cm	97.37 ± 9.05	101.04 ± 10.33	103.36 ± 11.73	104.22 ± 9.66	<0.001	105.49 ± 6.30	106.82 ± 7.81	109.02 ± 7.50	110.94 ± 8.14	<0.001
TC, mmol/L	6.09 ± 1.46	6.03 ± 1.25	6.15 ± 1.25	6.12 ± 1.41	0.885	5.92 ± 2.19	6.08 ± 1.38	5.89 ± 1.16	5.99 ± 1.41	0.409
LDL-C, mmol/L	3.83 ± 1.24	3.87 ± 1.07	3.96 ± 1.02	3.75 ± 1.14	0.4	3.48 ± 1.16	3.71 ± 1.01	3.60 ± 1.02	3.70 ± 1.25	0.825
HDL-C, mmol/L	1.41 ± 0.34	1.34 ± 0.27	1.32 ± 0.25	1.25 ± 0.29	0.004	1.01 ± 0.22	1.11 ± 0.25	1.04 ± 0.19	1.03 ± 0.23	0.211
TG, mmol/L	1.88 ± 1.21	1.92 ± 1.87	1.89 ± 0.89	2.42 ± 1.49	0.002	3.99 ± 9.17	2.63 ± 1.73	2.76 ± 2.10	2.86 ± 1.43	0.017
Plasma glucose, mmol/L	6.66 ± 2.00	6.65 ± 2.18	6.58 ± 1.47	6.76 ± 1.66	0.197	7.32 ± 2.62	6.37 ± 1.82	6.05 ± 0.55	6.41 ± 1.08	0.039
MetS components	3.66 ± 0.74	3.75 ± 0.75	3.78 ± 0.78	4.10 ± 0.79	<0.001	3.77 ± 0.80	3.74 ± 0.70	3.89 ± 0.79	4.15 ± 0.70	0.019
SBP, mmHg	134.60 ± 13.65	136.47 ± 15.96	141.07 ± 15.90	138.69 ± 15.16	0.022	137.87 ± 13.05	134.18 ± 13.15	137.02 ± 14.18	141.44 ± 14.48	0.036
DBP, mmHg	79.67 ± 8.82	80.39 ± 9.80	82.02 ± 8.60	81.56 ± 9.24	0.224	84.81 ± 8.48	84.14 ± 9.38	84.56 ± 10.83	87.18 ± 9.09	0.259

SUA—serum uric acid; BMI—body mass index; WC—waist circumference; TC—total cholesterol; LDL-C—low-density lipoprotein cholesterol; HDL-C—high-density lipoprotein cholesterol; TG—triglycerides; MetS—metabolic syndrome; SBP—systolic blood pressure; DBP—diastolic blood pressure.

**Table 2 medicina-58-00297-t002:** Prevalence of clusters of MetS components and SUA.

Risk Factors Combination	Men (*n* = 220)			Women (*n* = 386)		
	*n*	%	SUA, μmol/L	*n*	%	SUA, μmol/L
5 risks factors						
GHTBW	52	23.6	418.92 ± 84.49	90	23.3	373.12 ± 100.31
4 risks factors						
GTBW	42	19.1	436.76 ± 76.51	50	13.0	327.68 ± 77.73
GHBW	19	8.6	395.32 ± 92.73	52	13.5	342.90 ± 74.47
HTBW	13	5.9	413.92 ± 67.39	23	6.0	347.35 ± 69.92
GHTB	11	5.0	400.91 ± 75.49	8	2.1	303.00 ± 71.47
GHTW	6	2.7	371.00 ± 49.67	6	1.6	364.33 ± 65.56
3 risks factors						
GBW	37	16.8	384.00 ± 68.86	111	28.8	313.83 ± 67.57
TBW	13	5.9	411.31 ± 74.65	17	4.4	331.71 ± 63.30
HBW	6	2.7	408.83 ± 62.59	11	2.8	345.09 ± 76.04
GTB	11	5.0	368.91 ± 70.86	4	1.0	341.75 ± 64.86
GHW	2	0.9	366.50 ± 20.51	6	1.6	321.00 ± 67.36
GHB	4	1.8	318.50 ± 66.78	1	0.3	255.00
HTW	2	0.9	400.00 ± 46.67	3	0.8	304.67 ± 15.18
GTW	-	-	-	3	0.8	221.67 ± 76.14
HTB	2	0.9	408.00 ± 77.78	-	-	-
GHT	-	-	-	1	0.3	348.00

SUA—serum uric acid; G—elevated plasma glucose; H—low high density lipoprotein cholesterol; T—high triglycerides; B—elevated blood pressure; W—increased waist circumference.

**Table 3 medicina-58-00297-t003:** The odds ratio and *p*-Value of specific clusters of MetS components.

MetS Components Combination *	Crude		
	OR	95% C.I.	*p*-Value
5 risks			
GHTBW	1.982	1.351–2.906	<0.001
4 risks			
GHTB	0.475	0.156–1.450	0.191
GHTW	0.601	0.161–2.243	0.449
GTBW	0.927	0.585–1.469	0.748
GHBW	1.058	0.632–1.769	0.831
HTBW	1.030	0.511–2.076	0.935
3 risks			
GBW	0.653	0.436–0.978	0.039
GTB	0.655	0.206–2.082	0.473
HBW	1.282	0.481–3.418	0.619
TBW	1.056	0.493–2.262	0.889

* G—elevated plasma glucose; H—low high density lipoprotein cholesterol; T—high triglycerides; B—elevated blood pressure; W—increased waist circumference. All models adjusted for age and sex.

## Data Availability

Not applicable.

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
