# Peer review of "Which Clusters of Metabolic Syndrome Are the Most Associated with Serum Uric Acid?"

_medicina, 2022, doi:10.3390/medicina58020297_

Round 1
Reviewer 1 Report
The manuscript has substantially improved.
Author Response
Dear Reviewer,
Thank you for your time reviewing the article.
Kind regards,
Jurgita Mikolaitytė
Reviewer 2 Report
The work is interesting and clear writing, but there are some places have to be corrected.
The explanation of an abbreviation NCEP ATP III should be added in Abstract
In the Table 1 are presented results of MetS components‘. They are given in N, so it seems they should be a whole number without the decimal. It is not clear, why number of components are presented for example as 3,66±0,74 and so on.
The claim in 263 row should be based on literature sources.
Author Response
Dear Reviewer,
thank you for your comments. We have corrected the article accordingly.
Data in the article are expressed as mean ± standard deviation for continuous variables. This information is provided in the section Materials and Methods - in 78-88 rows. Table 1 shows the mean number of metabolic components for patients in the respective quartiles. We've clarified the name so it's not misleading.
Kind regards,
Jurgita Mikolaitytė
This manuscript is a resubmission of an earlier submission. The following is a list of the peer review reports and author responses from that submission.
Round 1
Reviewer 1 Report
The purpose of the present study may be of interest, although the paper is poorly written and not easy to read. Furthermore, the population taken into account and the methodology used may not be appropriate. These aspects likely contributed to the lack of clinical relevance of both results and conclusions.
In the manuscript, the authors compared cholesterol and blood pressure levels between the SUA quartiles. However, they did not specify anything regarding the lipid-lowering and anti-hypertensive therapy of these patients. The therapy could have affected the associations found making the work uninterpretable.
Although the sample analysed was large, the authors should clarify the reason for the clustering of the study population, making subgroups with a very low number of subjects. Then, they used these subgroups to assess the association with SUA levels (Figure 3 and Table 4). In my opinion, this process is not rigorous, because it compares too small groups to perform a logistic regression.
In table 4, the authors cited “hyperuricaemia” without giving any scientific definition. Reference MetS component / cluster was not specified. Furthermore, it is not clear why all analyses have been divided by gender, except for the most important analysis in Table 4.
Minor comments:
- In Figure 2, the authors should better specify which analysis has been used and among which groups.
Author Response
Thank you for your comments and remarks. We've updated the text, especially the description of the results section to reflect your comment on the clarity of the text.
The population in our research was used from the National primary prevention program (Lithuanian High Cardiovascular Risk primary prevention program (LitHir)). LitHir program is adopted in the whole country and detailed cardiac examination is performed only for patients with metabolic syndrome for detecting patients at high and very high cardiovascular risk. Our studied population was additionally tested for serum uric acid concentration. The clinical purpose is to help determine patients, which might be at very high risk, by the clusters of metabolic syndrome.
The association of hyperuricemia with combinations of components of the metabolic syndrome was tested for each combination individually with patients without these particular components of the metabolic syndrome, for example, for the 5 components, the reference group was patients without the 5 components.
For the data in Table 4, we used separate multivariable models, i.e., for each combination separately, and adjusted each for age and gender. This information is provided in the Material and Methods section. For the sake of clarity, we have provided this information in the results section next to Table 4. In addition, we included a definition of hyperuricemia.
We performed an analysis of the medication used by the subjects according to the uric acid quartiles. Angiotensin-converting enzymes, angiotensin II receptor blockers, calcium channel blockers, beta-adrenergic blocking agents, and statins were distributed among SUA quartiles without statistical significance. Only indapamide was statistically significant. However, we think that this did not influence the data concerning the clustering of components of metabolic syndrome.

Reviewer 2 Report
General comment
The Authors performed an interesting study evaluating the relationship between hyperuricemia and metabolic syndrome factors according to gender. The authors found a significant association in women but not in men.
I have some comments aimed at improving the quality of the manuscript.
Specific comment
The Authors do not report any data about liver e.g. aminotransferase levels, presence of fatty liver. The risk of hyperuricemia in patients with metabolic syndrome may be influenced by the presence of fatty liver. NAFLD is linked to both metabolic syndrome and hyperuricemia. Evidence suggests that serum uric acid is key for histopathogenesis of NAFLD and is linked to cardio-metabolic derangements (Ballestri S, Hepatol Res. 2016;46:1074-1087; Lonardo A, Dig Liver Dis. 2015;47:181-90; Mantovani A, Dig Liver Dis. 2018;50:518-520.). The Authors should add further analysis if possible based on available data. Please comment and update literature.
Author Response
Thank you for your comment and insight. No data on liver parameters were collected in this study. However, we do have plans to further investigate patients with metabolic syndrome participating in the National prevention program (Lithuanian High Cardiovascular Risk primary prevention program) in the future, based on liver function and other cardiometabolic indicators.
Round 2
Reviewer 1 Report
Although some points have been clarified, the weaknesses of the study remain.
Author Response
Dear Reviewer,
could you please specify the flaws of our study design?
Kind regards,
Jurgita Mikolaityte
Reviewer 2 Report
The Authors should add among study limitation the fact the they do not have data about liver (see previous comment).